# Person-centred care transitions for people with stroke: study protocol for a feasibility evaluation of codesigned care transition support

Maria Flink [1,2] Sebastian Lindblom,[1,2] Malin Tistad,[3] Ann Charlotte Laska,[4] Bo Christer Bertilsson,[1] Carmen Wärlinge,[5] Jan Hasselström,[1,5] Marie Elf [3] Lena von Koch [1,6] Charlotte Ytterberg[1,2]

¹Department of Neurobiology, Care Sciences and Society, Karolinska Institutet, Stockholm, Sweden
²Theme of Women's Health and Allied Health Professionals, Karolinska University Hospital, Stockholm, Sweden
³School of Health and Welfare, Dalarna University, Falun, Dalarna, Sweden
⁴Department of Clinical Sciences Danderyd Hospital, Karolinska Institutet, Stockholm, Sweden
⁵Academic Primary Health Care Centre, Region Stockholm, Stockholm, Sweden
⁶Theme Neuro, Karolinska University Hospital, Stockholm, Sweden

**Correspondence to**
Dr Maria Flink; maria.flink@ki.se

## ABSTRACT

**Background** Care transitions following stroke should be bridged with collaboration between hospital staff and home rehabilitation teams since well-coordinated transitions can reduce death and disability following a stroke. However, health services are delivered within organisational structures, rather than being based on patients' needs. The aim of this study protocol is to assess the feasibility, operationalised here as fidelity and acceptability, of a codesigned care transition support for people with stroke.

**Methods** This study protocol describes the evaluation of a feasibility study using a non-randomised controlled design. The codesigned care transition support includes patient information using videos, leaflets and teach back; what-matters-to me dialogue; a coordinated rehabilitation plan; bridged e-meeting; and a message system for cross-organisational collaboration. Patients with stroke, first time or recurrent, who are to be discharged home from hospital and referred to a rehabilitation team in primary healthcare for continued rehabilitation in the home will be included. One week after stroke, data will be collected on the primary outcome, namely satisfaction with the care transition support, and on the secondary outcome, namely health literacy and medication adherence. Data on use of healthcare will be obtained from a register of healthcare contacts. The outcomes of patients and significant others will be compared with matched controls from other geriatric stroke and acute stroke units, and with matched historic controls from a previous dataset at the intervention and control units. Data on acceptability and fidelity will be assessed through interviews and observations at the intervention units.

**Ethics and dissemination** Ethical approvals have been obtained from the Swedish Ethical Review Authority. The results will be published open-access in peer-reviewed journals. Dissemination also includes presentation at national and international conferences.

**Discussion** The care transition support addresses a poorly functioning part of care trajectories in current healthcare. The development of this codesigned care transition support has involved people with stroke, significant other, and healthcare professionals. Such involvement has the potential to better identify and reconceptualise problems, and incorporate user experiences.

### Strengths and limitations of this study

- ► The intervention was developed following the guidelines of the Medical Research Council.
- ► The intervention was developed using a codesign methodology with people with stroke, significant other and healthcare professionals.
- ► The use of mixed methods will provide knowledge regarding both process and outcomes.
- ► The study is non-randomised, but the intervention group will be compared to matched controls.

**Trial registration number** http://www.clinicaltrials.gov id: NCT02925871. Date of registration 6 October 2016.
**Protocol version** 1.

## BACKGROUND

Stroke, a leading causes of disability,[1] has an abrupt onset and is a stressful event, both for persons with stroke, and their families.[2] The past decades have seen improvements in medical and acute care of stroke as well as a reduction of the mean length of hospital stay in stroke units in high income countries.[3] Strong evidence and consensus have shown that stroke care should initially be supplied in hospital stroke units in the acute phase,[4] followed by a period of rehabilitation to regain functioning and to receive psychosocial support.[2] Thus, a care transition—a shift in responsibility from one healthcare setting to another—is almost always necessary. This may entail a care transition to inpatient rehabilitation, or directly to rehabilitation in the home provided by, for example, a multiprofessional team in primary healthcare .

Uncoordinated care transitions impose a burden on patients and their significant others, especially when they lack information to navigate the healthcare system.[2] The consequences of stroke such as aphasia, cognitive

impairment, poststroke fatigue and depression,[5] often render the stroke patients and their significant others unprepared for the care transition to the home.[6 7] This unpreparedness is further exacerbated by the sudden onset of stroke. In addition, a short hospital stay, little time to participate in transition planning,[8 9] and fragmented healthcare may contribute to patients' unrealistic expectations,[7] and their sense of being abandoned[10] in a new and complex life situation. Although Swedish laws and other regulations require care providers to coordinate care, a great responsibility is put on patients and significant others to coordinate their own care,[11 12] while few resources are offered to support a patient's ability to participate in their own care or to self-manage.[13 14]

Several care transition interventions have been developed, both for people with stroke and for other populations. Strong evidence supports the conclusion that, to reduce rehospitalisations, such care transitions must include both predischarge and postdischarge activities, and support for patients' capacity for self-management.[15 16] Such predischarge and postdischarge activities include discharge planning, structured discharge information, coordinated follow-up, and timely communication between care providers.[17 18] For people with stroke, calls have been made for self-management trials, as there is a lack of robust conclusions regarding effectiveness.[19] In stroke care transitions, extensive evidence shows that the rehabilitation care transition Early Supported Discharge (ESD) can reduce death and disability after a mild to moderate stroke.[20] ESD implies that a stroke-specific, multidisciplinary team plans and coordinates hospital discharge, and provides rehabilitation in the community context.[21] Despite the evidence in its favour, ESD has scarcely been implemented in Sweden. A possible explanation for the lack of implementation is that cross-organisational health services, like ESD, may be difficult to accomplish in fragmented healthcare systems with firm borders between hospital and community care.

There is hence a need for a rehabilitation care transition that consider patient's needs and capacities, as well as organisational settings. This need was highlighted in our prospective observational study, conducted between April 2016 and February 2018, in Stockholm, Sweden[22 23] with 206 people with acute stroke, of the current state of transitions from stroke and geriatric hospital units to the home with subsequent rehabilitation in primary healthcare. One week after hospital discharge, participants were followed up regarding satisfaction with care transition; and then again, at 3 and 12 months, regarding functioning in everyday life and use of healthcare. In addition, we performed a grounded theory study of perceptions of current care transitions of patients, families and healthcare professionals. The findings from the prestudies[22 23] identified a need for improved dialogue between patients and professionals; between professionals working in the same care provider organisation; and between professionals in different organisations. The improved dialogue means a communication where all parties communicate

on equal terms and strive for a shared understanding. Further, a need to enhance patient preparedness for discharge, and self-management at home was identified. Based on the results,[22 23] we identified a need to develop bridging communicative links during the transition process, both within and between care provider organisations, as well as between patients and healthcare professionals. These links aim to ensure a coordinated rehabilitation transition, and to ensure that patients and their significant others feel prepared for home coming.

### Development of the intervention

The intervention is hence based both on our prestudies[22 23] and the stroke care trajectory literature,[4 20] and was developed in a two-step codesign process.[24]

First, a trained facilitator held five codesign workshops with people with stroke, one significant other, and healthcare professionals from hospital stroke units and multidisciplinary home rehabilitation teams. Details of the codesign process and the participants have been presented previously.[25] Three areas of unmet needs related to care transitions were identified across patients, significant others, and healthcare professionals:

1. *Shared understanding of patient illness and situation.* Participants identified a need for: repeated information about the present situation and expected future; communication adapted to the situation; updated written information; time to ask questions; structured and timely information.
2. *Preparedness for homecoming.* Participants identified a need for: feeling safe and understood at the discharge; clarity about services provided by the home rehabilitation team; preparedness for homecoming; clarity about medications; their perspectives and needs to be considered.
3. *Coordination.* Participants identified a need for: coordination between stroke unit and home rehabilitation teams; clarity about care trajectory; collaboration between sites.

Based on these needs, suggestions for improvements were codesigned by participants.

Thereafter, interactive e-meetings with healthcare professionals were held in which suggestions from the codesign workshops were further developed and contextually adapted into the intervention.

### The intervention: the codesigned care transition support

The intervention consists of several elements that aim to meet patients', significant others', and professionals' needs for shared understanding, patient preparedness for homecoming, and coordination (figure 1).

Three different sources of home-rehabilitation information will be offered to patients and significant others: *two informational videos and one pamphlet*. First, an informational video including general information about home rehabilitation will be shown during daytime in the day room. Second, an informational video featuring professionals from the patient's specific home-rehabilitation team will be

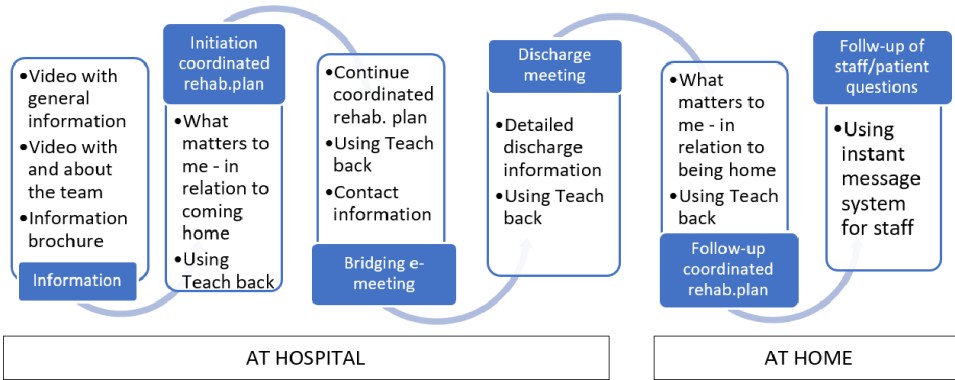

**Figure 1** The codesigned care transition support.

provided to patients on digital tablets. Third, an informational pamphlet including general information about home rehabilitation, a link to the information video, and contact information to the specific team, will be provided to patients. After the patient has accessed these sources of information, hospital staff will initiate a dialogue with the patient about '*what matters to me*'in relation to coming home and continued rehabilitation. In this dialogue, hospital staff and the patient will initiate the development of a *coordinated rehabilitation plan* based on the patient's needs and wishes for continued rehabilitation at home, as well as hospital staffs' assessment. Based on agreements in the initiated plan, the hospital staff will refer the patient to the home-rehabilitation team and book an e-meeting with the team before discharge. This *bridged e-meeting* aims to connect the patient with the team that will be responsible for rehabilitation at home through a first personal contact and decide an appointment for the first home visit. The e-meetings will include a dialogue including the patient, the significant other, the hospital staff and the rehabilitation team. The dialogue will include the patient's situation, needs, wishes and fears in relation to coming home; a presentation from the home rehabilitation team about their services; a possibility for the patient and significant other to ask questions, and to continue the initiated coordinated rehabilitation plan for the homecoming. The e-meeting will function as an in-reaching bridge connecting patient, hospital staff, and home rehabilitation team. After homecoming, the home rehabilitation team, the patient, and significant others will review and revise the initiated coordinated rehabilitation plan.

*Teach back*[26 27] is a person-centred, iterative communication method focused on a shared understanding of the patient's situation and the professional's information. The aim of *Teach back* is both to ensure that health professionals have understood the patient correctly, and to improve patient awareness of the own health condition, medications, and how to navigate the healthcare system, that is, improved health literacy. Teach back will be used in the *'what matters to me'* dialogue, the bridged e-meeting, and discharge encounters. The professionals will provide information and thereafter ask the patient to repeat in their own words what was said, emphasising that this is to ensure that professionals explained satisfactorily. The detailed discharge information entails information about follow-up, what to consider after discharge, and where to turn with questions.

A safe *instant messaging system,* within the electronic health record, will ensure interorganisational communication among healthcare professionals. This will ensure the exchange of information and resolve remaining questions regarding patient care between the hospital and home-rehabilitation, such as a revoked driving license or medications.

The codesigned care transition support is grounded in principals of person-centred care[28] and of integrated care[29] on a microlevel and mesolevel. This entails that the development of the intervention, and the intervention itself, are both based on individuals' preferences, needs, and values, taking patient resources into consideration, that is, the intervention is person centred,[28] and strives to be realistic regarding what healthcare professionals in the included organisational settings can provide. We adhere to the person-centred 'Rainbow model of integrated care'.[29] This model emphasises the need for both functional and normative links between patients and professionals, as well as within and between hospital and primary care organisations. Functional here refers to the information systems that link the hospital and primary care organisations. Normative here refers to shared understanding and values, both within and between organisations.

### Care transitions in control group

Our previous studies identified that a rehabilitation care transition 'as usual' is initiated by an electronic referral from hospital healthcare professionals to the receiving neurorehabilitation team. The referral notifies the neurorehabilitation team about the patient and of the discharge. The neurorehabilitation team is obliged to initiate contact with the patient within 48 hours of referral and/or hospital discharge.

### Aims and hypothesis

The primary aim is to assess the feasibility, operationalised as fidelity and acceptability, of a codesigned care transition support for people with stroke.

The following research questions (RQ) will be used:

**RQ1**: Is the intervention delivered as intended (fidelity)?

**RQ2**: Is the intervention acceptable in terms of content and delivery?

**RQ3**: Do patient and significant other outcomes differ between patients who received the intervention and matched controls, or between intervention patients and matched historic controls?

We hypothesise that information provided in a person-centred dialogue, as well as a discharge focused on bridging the gap between hospitalisation and home, will help patients and significant others feel more satisfied with the transition, and be better prepared for home-coming. This may, in turn, help patients regain functioning and reduce healthcare utilisation.

## METHODS

This study protocol describes the evaluation of feasibility for the codesigned care transition support using a non-randomised controlled design. Updates on the study protocol will be published on clinical trials.

The project adheres to the framework for development and evaluation of complex interventions by the Medical Research Council, UK,[30] which calls for phased and iterative approaches in the design and evaluation of a complex intervention. This project hence draws on our prestudies of the perceptions, satisfaction and outcomes of current state of care transitions, in which we have involved patients, significant others, and healthcare professionals in several care trajectories.[22 23]

### Patient and public involvement

This study protocol for a feasibility study includes a description of the development of the intervention. The intervention was developed using a codesign methodology that involved five workshops, followed by continuous discussions with people with stroke, a significant other, and healthcare professionals from hospital and primary care rehabilitation. The codesign included joint development of the interventions' components, contextual factors to consider, participant needs, and important outcomes to target. Details of the codesign process and the participants have been presented previously.[25]

### Study design

This is a feasibility study, with a non-randomised controlled study design. The intervention will be implemented at a geriatric stroke unit and an acute stroke unit in Hospital A in Stockholm, Sweden. The controls will be recruited from a geriatric stroke unit and an acute stroke unit at Hospital B in Stockholm, Sweden. Thereto, our previous studies at hospitals A and B (prestudy conducted from 2016 to 2018) provide historic controls using predata from the same hospital units. This design will allow us to compare intervention patients with matched controls, intervention patients with matched historic controls, and control patients with historic controls. Controls will be matched according to age, gender, stroke severity and civil status (living alone or cohabiting).

## Participants
### Patients

Inclusion criteria: patients who have had a first time or recurrent stroke, and who will be discharged home from the participating stroke units and referred to a rehabilitation team in primary healthcare for continued rehabilitation in the home. Exclusion criteria: patients unable to give informed consent, due to for example, severe aphasia (National Institutes of Health Stroke Scale (NIHSS)[31] language/communication 2 or 3) or dementia. Most patients discharged home from these units have mild to moderate stroke.[28] The patients will be informed of the study and invited to participate at the hospital by a research assistant. The research assistant will provide oral and written information about the study and obtain consent to participate.

### Sample size

In total, 50 persons will be consecutively included, 25 from the intervention site and 25 from the control site.

### Significant others

Significant others will be included via the included patients. The included patients will be asked to name a significant other to be invited to also participate in the study. The significant others will be mailed written information about the study including an informed consent and a prestamped envelope. Significant others who return signed consent will be included in the study. Patients who do not have, or do not wish to name a significant other, will remain included.

### Healthcare professionals

Hospital staff and home rehabilitation team members who have delivered the intervention will be invited to participate in the study. A purposive sampling (in terms of profession, years in profession) will be used. Approximately 15 hospital and home rehabilitation professionals will be included.

### Data collection
#### RQ1 and RQ2

Data on acceptability and fidelity of the intervention will be collected by participant observations, interviews, and data from the healthcare record. At the intervention site, observations will be conducted of the 'what matters to me' dialogue, the bridged e-meeting, and discharge encounters. The observations will be made using a structured protocol in which the following criteria will be assessed: use of plain language, open-ended questions vs yes/no questions, asking patients to repeat information in their own words, repeating/revising information as needed. Individual interviews with the participating healthcare professionals and patients will be conducted after these observations. Interviews with healthcare professionals will target perspectives on patient understanding of information; the professionals' understanding of patient needs, resources, and will; experiences of collaboration with patients and healthcare professionals;

and experiences of the 'what matters to me' dialogue, the bridged e-meeting, and discharge encounters. Patient interviews will target perspectives on their understanding of information material; how healthcare professionals explained information; experiences of collaboration with healthcare professionals; and how their perspectives and resources were considered.

After the 25 patients have been discharged with the codesigned care transition support, focus group interviews of healthcare professionals at the intervention site will be conducted regarding their overall understanding of the intervention (including the training), their experience delivering the intervention, and their perspectives on the content of the intervention. Patients and significant others at both the intervention and the control sites will be individually interviewed on their experiences of communication with healthcare professionals at hospital, how their needs were taken into consideration, and their satisfaction with and preparedness for discharge and homecoming. All interviews will be audio recorded and transcribed verbatim.

Data on number of e-meetings and instant messages, and the content of the coordinated rehabilitation plan, will be collected from the healthcare records of the included patients.

### RQ3

Data on likely effectiveness will be collected using questionnaires and registry data.

Baseline data on participant characteristics, length of hospital stay, disease-related data, and functioning in everyday life will be retrieved from medical records. Data on our primary outcome, satisfaction with codesigned care transition support, will be collected with the Care Transition Measure.[32 33] Data on our secondary outcome, health literacy, will be collected with the Health Literacy Questionnaire.[34] These questionnaires together with an envelope (addressed and postage paid), will be sent by mail within 1 week of discharge from the hospital.

Data on the outcomes of patients and significant others at 3 months after stroke will be collected using questionnaires in structured interviews. Data will be collected from people with stroke regarding health literacy,[34] medication adherence,[35] cognitive function,[36] signs of anxiety and depression,[37] fatigue,[38] perceived impact of stroke on function, activities and participation in everyday life,[39] need of assistance in performing personal activities of daily living,[40] participation in complex and social activities,[41] health-related quality of life[42] and self-efficacy.[43] Data on use of healthcare will be obtained from the register of healthcare contacts in the Region Stockholm. Data on caregiver burden,[44] and informal care supplied will be collected using questionnaires in structured interviews with significant others. The same reliable and validated measures and data collection timepoints have been applied in the prestudy of patient and significant other outcomes in current care transitions.

### Analyses

Observations and interviews will be analysed using qualitative content analysis.[45] Qualitative content analysis is a method well suited for descriptive qualitative analysis.[45] Likely effectiveness of the codesigned care transition support (RQ3) in terms of outcomes and resource use, will be examined statistically and compared with control and prestudy historic data. The statistician performing this analysis will be blinded to group allocation.

### Ethics and dissemination

Ethical approvals have been obtained from the Swedish Ethical Review Authority, nos. 2015/1923-31/2, 2019-04219, and 2021-02274. The proposed project will be conducted in accordance with the Declaration of Helsinki; oral and written consent to participate will be obtained from all participants. All study-related information about participants will be stored securely at the Karolinska Institutet. Only the researchers within the group will have access to data.

The results will be published open-access in peer-reviewed journals. Dissemination also includes presentation at national and international conferences.

### DISCUSSION

The primary aim is to assess the feasibility, operationalised as fidelity and acceptability, of the codesigned care transition support for people with stroke. The codesigned care transition support will address a poorly functioning part of the care trajectories in current healthcare, as identified in our prestudies[22 23] and the stroke care trajectory literature.[4 20] A recent Cochrane review identified that discharge planning studies mainly focused on individualised discharge plans in order to reduce readmissions.[18] Such plans may render a small reduction in the number of hospital readmissions, but the development of care transitions has so far not involved patients or significant others, nor all the professionals among whom coordination is required. The development of this codesigned care transition support has therefore involved all these users.[46] Such involvement has the potential to better identify and reconceptualise problems and incorporate user experiences.[46]

**Contributors** MF, SL, MT, ACL, ME, LvK and CY drafted the manuscript. BCB, CW, and JH made significant contributions in revising and rewriting the manuscript. All authors have read and approved the manuscript.

**Funding** This work was supported by the Doctoral School in Healthcare Sciences [2-134/2016], the Kamprad Family Foundation grant number [20190185], Neuro Sweden, and the Swedish Stroke Association.

**Disclaimer** The funders neither had, nor will have, any role in design, data collection, data analysis, interpretation of data, writing of manuscripts or in decision to submit the paper for publication.

**Competing interests** None declared.

**Patient and public involvement** Patients and/or the public were involved in the design, or conduct, or reporting, or dissemination plans of this research. Refer to the Methods section for further details.

**Patient consent for publication** Not applicable.

**Provenance and peer review** Not commissioned; externally peer reviewed.

**ORCID iDs**
Maria Flink http://orcid.org/0000-0003-0536-0024
Marie Elf http://orcid.org/0000-0001-7044-8896
Lena von Koch http://orcid.org/0000-0002-8560-3016

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
