## [Reviewer comments · BMJ Open]

ARTICLE DETAILS

TITLE (PROVISIONAL)	Person-centred care transitions for people with stroke: Study protocol for a feasibility evaluation of co-designed care transition support
AUTHORS	Flink, Maria; Lindblom, Sebastian; Tistad, Malin; Laska, Ann Charlotte; Bertilsson, Bo Christer; Wärlinge, Carmen; Hasselström, Jan; Elf, Marie; von Koch, Lena; Ytterberg, Charlotte

VERSION 1 – REVIEW

REVIEWER	Sarah Tyson University of Manchester, Stroke & Vascular Research Centre, School of Nursing, Midwifery & Social Work
REVIEW RETURNED	25-Jan-2021

GENERAL COMMENTS	This a well written and structured paper presenting the protocol for a feasibility study of a hospital discharge support/preparation package for people with stroke who will be receiving on-going rehabilitation once discharged. This is an important and much neglected area of rehab, so it is a high impact and timely topic. The intervention has been co-produced with stroke survivors, their 'significant others' and HCPs. It is great to see co-production being used. The authors have done a lot of work to develop this intervention. I look forward to seeing the results of this study and learning more about the intervention. A few thoughts Abstract  • One of the aims is said to be 'to assess likely effectiveness'. This filled me with dread and I feared I was going to be reading yet another underpowered trial making claims of efficacy. This does not appear to be the case. In fact, the authors appear to be mainly using the outcome data to generate a sample size calculation for a future trial- which is fine (and a relief!) so it would be clearer to say so. If wanting to dabble in notions of outcome, then the most the authors could say was that they aimed to compare outcomes in the different groups. But one CAN NOT INFER EFFICACY ((or effectiveness) from a cohort study. • The study design is referred to as a "Non-randomised controlled design", which is a rather unusual way to describe a controlled cohort study. It isn't wrong but a tad vague and uninformative. • The term 'rehabilitation link' is used to describe the intervention. This isn't one I have come across previously and it isn't clear what is being linked. As far as I can tell, what is described is a discharge/transition support package, a term which is widely used. It might be worth changing the name either to make clear what is being linked (presumably hospital and community rehabilitation) or call it a discharge support package. • No details of selection criteria or sample size in the method section.
--

	Introduction In the section describing the content of the invention, ‘the professional’ is often referred to but it is not clear whether this is the hospital based or community based professional – which is important to clarify. It appears that the rehabilitation plan is devised by a hospital based HCP and the patient for the community based team to deliver. Is this correct? Such an arrangement would not be well received in the UK! Community teams would expect to be involved in devising any treatment plan they were expected to deliver. They would be concerned that hospital staff and/or patient would make plans which they were unable to deliver, given that hospital staff cannot have a detailed understanding of the day-to-day context in which the community teams are working. But then perhaps staffing shortages and under-resourcing are less of a problem in Sweden than the UK Method  • How will controls be matched regarding age and stroke severity? How far back will they authors go in the ‘past patients’ to find ones who match the patients receiving the intervention? • Will recruitment be restricted to those who can give full consent themselves? Are there any arrangements for those who can’t consent? If not, a large and important group of stroke survivors will be excluded. I would have thought it important to test the feasibility and acceptability of this intervention in all people for whom it may be, eventually, used. Thus I urge the authors to consider how the selection criteria could be expanded to include all patients who need to receive on-going rehabilitation. If that isn’t possible it needs to be acknowledged as a limitation. • It would be good to clarify whether the sample size calculation is for a future cluster RCT (ie all patients will receive the intervention), or are the authors attempting to just provide the link intervention for people who are recruited? • Good to see patient satisfaction as the primary outcome - rarely done, and it should be more frequent!
--	---

REVIEWER	Molly Manning University of Limerick
REVIEW RETURNED	19-Mar-2021

GENERAL COMMENTS	BMJ Open March 2021 Thank you for the opportunity to review this important, interesting, and novel protocol to examine feasibility of a co-designed intervention to improve transitions across care settings. I believe this study will make a valuable contribution to the stroke literature and look forward to reading about your findings as it progresses. I have some comments and questions in relation to the protocol inclusion criteria and process. In the main, I would like to know how people with post-stroke communication impairment (including aphasia) will be supported to participate in the study, particularly in informed consent processes and in the qualitative component? What adaptations will be made, if any, to equalize their participation? If none, I would consider making an appropriate statement, particularly given the focus on improving dialogue and communication on equal terms across patients and care providers (P6). Secondly, I have several more minor comments and suggestions,
---

detailed below. Finally, the protocol would benefit from close editing throughout to address syntactic issues (e.g., “and a fragmented healthcare may contribute to unrealistic expectations”).

I hope you find the feedback constructive and would be happy to review a revised manuscript.

Abstract:

- “The primary aim is to assess the feasibility operationalized as fidelity, acceptability, and the secondary aim is to assess likely effectiveness of a co-designed rehabilitation link for persons with stroke”
- This may benefit from some minor editing and possibly splitting into 2 sentences to make more succinct, clearer.
- Transition from ‘transitions’ to ‘link’ – unclear if using interchangeably at this point

Strengths and limitations of this study

- Full stops.
- “The development of the intervention was conducted” could be shortened.
- Points 3-4 could be further developed to explicate the relative advantages of each approach.

Background

- “Psycho-social” – please review hyphen use.
- “Cognitive impairments, post-stroke fatigue and depression[5], as well as the sudden onset, often render the person having had the stroke and their significant others unprepared for the care transition to home[6, 7].” Suggest you reference post-stroke communication impairments, including aphasia here. Aphasia is risk factor for substandard care, including at transitions of care.
- L44-45 – unclear whose expectations are unrealistic.
- L47 - “Although laws and other regulations require care providers” – this appears to reference Swedish law so might be good to make this clear and/or to widen the focus to international stroke policy context.
- P5: “However, these interventions are not specifically directed to care transitions within rehabilitation or for people with stroke whose capacity for selfmanagement may be hindered by the cognitive and communicative impairments of the stroke.”
 - This statement is contentious and unsupported. Please adjust accordingly.
 - Please reference recent body of literature around shortcomings in including people with communication impairment post-stroke in self-

	management literature (e.g. work of Dr Faye Wray & colleagues), and systematic exclusion of people with aphasia across stroke literature more generally (e.g. work of Prof Marian Brady & colleagues).  • PL L15-21 – flow of information from self-management to ESD needs a little re-working. Development of the intervention  - P6, L41 “The participants identified several unmet needs of patients, significant others, and professionals within three areas of the care transition”: a little unclear. Suggest something along the lines of “3 areas of need around care transitions were identified across patient, significant other and professional participant groups”. Study design  ○ P11 L32, “Controls will be matched regarding age, gender, stroke severity and civil status (living alone or co-habiting).” Suggest “matched according to...” The intervention  ○ Please provide a little more detail about Rainbow model and/or enhance clarity of final 2 sentences in this section. Participants  ○ This section could be better structured better. Current use of headings and inclusion of sample size in its present format is unclear. ○ First line “patients and significant others” – however, no detail concerning significant others in the following paragraph. ○ Sample size: First sentence repetitious. Slightly confusing in the scope of content (e.g. significant other info / recruitment / sampling). ○ Healthcare professionals – description needs to be clearer. Care transitions in control group  ○ Break into >1 sentence. Ethics and dissemination  ○ “will be conducted in accordance with...” ○ “oral and written consent to participate will be...” (and delete “upcoming data collections”)
--	---

	Analyses  - Is 'targeting' the right word? - Can you provide additional detail around you content analysis approach? - 2nd sentence is unclear. Suggest something like: "Likely effectiveness of the co-designed rehabilitation link (RQ3) in terms of outcomes and resource use, will be examined statistically and compared with control and pre-study historic data." - Change 3rd sentence to make cleared, e.g. "Researchers will be blinded to participant allocation status when conducting statistical analyses". Discussion Content is appropriate, however close proof-reading is recommended.
--	---

VERSION 1 – AUTHOR RESPONSE

Reviewer 1	Our response	Page (P) and line (L)
Abstract		
One of the aims is said to be 'to assess likely effectiveness'. This filled me with dread and I feared I was going to be reading yet another underpowered trial making claims of efficacy. This does not appear to be the case. In fact, the authors appear to be mainly using the outcome data to generate a sample size calculation for a future trial- which is fine (and a relief!) so it would be clearer to say so. If wanting to dabble in notions of outcome, then the most the authors could say was that they aimed to compare outcomes in the different groups. But one CAN NOT INFER EFFICACY ((or effectiveness) from a cohort study.	We agree and have revised the aim so that it reads: "The primary aim is to assess the feasibility operationalized as fidelity and acceptability of a co-designed care transition support for persons with stroke."	P2 L2-3

The study design is referred to as a “Non-randomised controlled design”, which is a rather unusual way to describe a controlled cohort study. It isn’t wrong but a tad vague and uninformative.	We can understand that this is vague but have decided to keep this description as we believe this describes our design	
The term ‘rehabilitation link’ is used to describe the intervention. This isn’t one I have come across previously and it isn’t clear what is being linked. As far as I can tell, what is described is a discharge/transition support package, a term which is widely used. It might be worth changing the name either to make clear what is being linked (presumably hospital and community rehabilitation) or call it a discharge support package.	We can see that this is confusing and have revised it to “care transition support”	Throughout the manuscript
No details of selection criteria or sample size in the method section	We have added this information.	P2 L9-12
Introduction		
In the section describing the content of the invention, ‘the professional’ is often referred to but it is not clear whether this is the hospital based or community based professional – which is important to clarify.	This has been revised to improve clarity.	P7-8

It appears that the rehabilitation plan is devised by a hospital based HCP and the patient for the community based team to deliver. Is this correct? Such an arrangement would not be well received in the UK! Community teams would expect to be involved in devising any treatment plan they were expected to deliver. They would be concerned that hospital staff and/or patient would make plans which they were unable to deliver, given that hospital staff cannot have a detailed understanding of the day-to-day context in which the community teams are working. But then perhaps staffing shortages and under-resourcing are less of a problem in Sweden than the UK	We agree – the text is misleading. We have added information that the home-rehabilitation team, the patient and significant other will review and revise the plan after the home coming.	P8 L9-10
Method		
How will controls be matched regarding age and stroke severity? How far back will they authors go in the 'past patients' to find ones who match the patients receiving the intervention?	We have added information that the historic controls refer to patients who were included in our pre-studies spanning from April 2016 to February 2018.	P11 L19-20
Will recruitment be restricted to those who can give full consent themselves? Are there any arrangements for those who can't consent? If not, a large and important group of stroke survivors will be excluded. I would have thought it important to test the feasibility and acceptability of this intervention in all people for whom it may be, eventually, used. Thus I urge the authors to consider how the selection criteria could be expanded to include all patients who need to receive on-going rehabilitation. If that isn't possible it needs to be acknowledged as a limitation.	In Sweden, most of the patients with stroke who are discharged from the hospital stroke units or geriatric stroke units to their homes can give consent themselves, based on the experiences from our pre-studies. The persons who need more rehabilitation are not discharged directly to their homes from these units. Thus, all patients discharged from the involved units to their homes are eligible. We have added information on this.	P11 L16-18

It would be good to clarify whether the sample size calculation is for a future cluster RCT (ie all patients will receive the intervention), or are the authors attempting to just provide the link intervention for people who are recruited?	The statement “In total 50 persons will be consecutively included, 25 patients from the intervention site and 25 patients from the control site will be recruited” refers to the number of patients to be recruited in the feasibility study. The intervention will be provided to these 25 persons.	
Good to see patient satisfaction as the primary outcome - rarely done, and it should be more frequent!	Thank you. We agree!	

Reviewer 2	Our response	Line and pages
Abstract		
“The primary aim is to assess the feasibility operationalized as fidelity, acceptability, and the secondary aim is to assess likely effectiveness of a co-designed rehabilitation link for persons with stroke” This may benefit from some minor editing and possibly splitting into 2 sentences to make more succinct, clearer.	We agree and have revised the aim so that it reads: “The primary aim is to assess the feasibility operationalized as fidelity, and acceptability of a co-designed care transition support for persons with stroke.”	P2 L2-3
Transition from ‘transitions’ to ‘link’ – unclear if using interchangeably at this point	We have revised the term “rehabilitation link” to “care transition support” to better reflect the proposed intervention.	Throughout the manuscript
Strengths and limitations of this study		
“The development of the intervention was conducted” could be shortened.	It has been revised and now reads: “The intervention was developed using co-design methodology with people with stroke, significant other and health care professionals”	Not in manuscript
Points 3-4 could be further developed to explicate the relative advantages of each approach.	We have revised these bullets points.	Not in manuscript

Background		
“Psycho-social” – please review hyphen use.	We have removed it.	P3 L17
“Cognitive impairments, post-stroke fatigue and depression[5], as well as the sudden onset, often render the person having had the stroke and their significant others unprepared for the care transition to home[6, 7].” Suggest you reference post-stroke communication impairments, including aphasia here. Aphasia is risk factor for substandard care, including at transitions of care.	Thank you. This has been added.	P3 L25
L44-45 – unclear whose expectations are unrealistic.	We have added “patients having”	P4 L3
L47 - “Although laws and other regulations require care providers” – this appears to reference Swedish law so might be good to make this clear and/or to widen the focus to international stroke policy context.	Thank you for noticing this, we have added that this refers to Swedish laws and regulations.	P4 L5
P5: “However, these interventions are not specifically directed to care transitions within rehabilitation or for people with stroke whose capacity for selfmanagement may be hindered by the cognitive and communicative impairments of the stroke.” o This statement is contentious and unsupported. Please adjust accordingly. o Please reference recent body of literature around shortcomings in including people with communication impairment post-stroke in self-management literature (e.g. work of Dr Faye Wray & colleagues), and systematic exclusion of people with aphasia across stroke	Thank you for noticing this lack of clarity. We have decided to delete this sentence to focus the information on what kinds of care transition studies that previously have been conducted.	P4 L18-21

literature more generally (e.g. work of Prof Marian Brady & colleagues).		
PL L15-21 – flow of information from self-management to ESD needs a little re-working	We have revised this paragraph.	P4 L18-21
Development of the intervention		
P6, L41 “The participants identified several unmet needs of patients, significant others, and professionals within three areas of the care transition”: a little unclear. Suggest something along the lines of “3 areas of need around care transitions were identified across patient, significant other and professional participant groups”.	Thank you for providing us with this friendly support. We have revised according to your suggestion.	P6 L8-10
Study design		
P11 L32, “Controls will be matched regarding age, gender, stroke severity and civil status (living alone or co-habiting).” Suggest “matched according to...”	Thank you for this suggestion. We have followed your advice.	P11 L24
The intervention		
Please provide a little more detail about Rainbow model and/or enhance clarity of final 2 sentences in this section.	Thank you for noticing this lack of clarity. We have revised these 2 sentences.	P9 L7-12
Participants		
This section could be better structured better. Current use of headings and inclusion of sample size in its present format is unclear.	We have reviewed this section and re-structured the headings.	P12-13
First line “patients and significant others” – however, no detail concerning significant others in the following paragraph.	We have added information on significant others inclusion process.	P12 L10-16
Sample size: First sentence	We have restructured this	P12-13

repetitious. Slightly confusing in the scope of content (e.g. significant other info / recruitment / sampling).	section, including headings, and provided more information about recruitment of significant others.	
Healthcare professionals – description needs to be clearer.	We have clarified and revised this section.	P12-13
Care transitions in control group		
Break into >1 sentence.	This has been revised.	P9 L20-25
Ethics and dissemination		
“will be conducted in accordance with...”	This has been revised.	P16 L4
“oral and written consent to participate will be...” (and delete “upcoming data collections”)	This has been revised.	P16 L5
Analyses		
Is ‘targeting’ the right word?	This has been revised.	P15 L18
Can you provide additional detail around your content analysis approach?	We have deleted the word “manifest” and instead added information that the method is well-suited for descriptive qualitative analysis.	P15 L19-20
2nd sentence is unclear. Suggest something like: “Likely effectiveness of the co-designed rehabilitation link (RQ3) in terms of outcomes and resource use, will be examined statistically and compared with control and pre-study historic data.”	Thank you for providing us with this suggestion. We have revised accordingly.	P15 L21-23
Change 3rd sentence to make clearer, e.g. “Researchers will be blinded to participant allocation status when conducting statistical analyses”	We have revised this so as it reads The statistician performing analysis will be blinded to group allocation.	P16 L1-2
Discussion		
Content is appropriate, however close proof-reading is recommended	Thank you for encouraging this.	

VERSION 2 – REVIEW

REVIEWER	Sarah Tyson University of Manchester, Stroke & Vascular Research Centre, School of Nursing, Midwifery & Social Work
REVIEW RETURNED	25-May-2021

GENERAL COMMENTS	The manuscript is now clear, thorough and robust. I look forward to hearing the results in due course.
--

REVIEWER	Molly Manning University of Limerick
REVIEW RETURNED	01-Jun-2021

GENERAL COMMENTS	BMJ Open June 2021 Thank you for the opportunity to re-review this paper, which is substantially improved. I have a few queries / suggestions below. I hope you find the feedback constructive. Supplemental file Please review Spirit table formatting / margins. Abstract:  • Content broadly clear, but there are grammatical issues to be addressed throughout (and throughout manuscript more generally). • Suggest specifying who the intervention was co-designed with. • Suggest expanding the discussion which is rather brief. • Similarly the study strengths section could be developed a little (e.g., why is it good that MRC was followed; or that a co-design approach was taken?). Background:  • P6, L4 “impose a burden on”. • P6 L6-8 – could be clearer. • P6 L9 – drop the “a” from “fragmented healthcare”. • P6 L15 – suggest you reference the treatment burden literature which would complement this content at international level e.g., GALLACHER, K., MORRISON, D., JANI, B., MACDONALD, S., MAY, C. R., MONTORI, V. M., ERWIN, P. J., BATTY, G. D., ETON, D. T., LANGHORNE, P. & MAIR, F. S. 2013. Uncovering treatment burden as a key concept for stroke care: a systematic review of qualitative research. PLoS Med, 10, e1001473. • You might also briefly mention the lack of effectiveness data around self-management interventions for stroke care more generally (e.g., WRAY, F., CLARKE, D. & FORSTER, A.
---

	2018. Post-stroke self-management interventions: a systematic review of effectiveness and investigation of the inclusion of stroke survivors with aphasia. Disability and Rehabilitation, 40, 1237-1251.)  • P5 L16-18 – could be clearer (“the improved dialogue includes a communication...”) Development of the intervention  - Some brief details about the co-designers with stroke would be welcome. This has implications for interpreting the unmet needs identified. - The 3 areas of unmet need are a little confusing as they each contain a paragraph of content. Could this be more succinct? - It is a little unclear how these sessions with people with strokes were co-design sessions as opposed to focus group discussions? It appears that the healthcare professionals were more involved in developing the intervention. Aims and Hypothesis  ○ P10 L12 – sentence could be split and made clearer. Public and Patient Involvement  ○ Again, it will need to be made clearer earlier in the paper how this was in fact co-design. Participants  ○ Once again, you need to detail inclusionary criteria. Will people with aphasia be included, and if so, how will informed consent be obtained? (Similarly, later on, process for conducting communicatively accessible interviews will need to be detailed if so).
--	--

VERSION 2 – AUTHOR RESPONSE

Reviewer 1:

Reviewer: The manuscript is now clear, thorough and robust. I look forward to hearing the results in due course.

Response: Thank you for the kind words.

Reviewer 2:

Reviewer: Please review Spirit table formatting / margins.

Response: The margins have been revised.

Reviewer: Content broadly clear, but there are grammatical issues to be addressed throughout (and throughout manuscript more generally)

Response: The manuscript has now been revised following the advice of a professional language editor.

Reviewer: Suggest specifying who the intervention was co-designed with

Response: We have added this information in the abstract.

Reviewer: Suggest expanding the discussion which is rather brief.

Response: We have expanded the discussion.

Reviewer: Similarly, the study strengths section could be developed a little (e.g., why is it good that MRC was followed; or that a co-design approach was taken?)

Response: We have revised the strengths and limitations section.

Reviewer: • P6, L4 “impose a burden on”. • P6 L6-8 – could be clearer. • P6 L9 – drop the “a” from “fragmented healthcare”

Response: We have revised following your advice.

Reviewer: P6 L15 – suggest you reference the treatment burden literature which would complement this content at international level

Response: We have revised following your advice.

Reviewer: You might also briefly mention the lack of effectiveness data around self-management interventions for stroke care more generally (e.g., WRAY, F., CLARKE, D. & FORSTER, A. 2018. Poststroke self-management interventions: a systematic review of effectiveness and investigation of the inclusion of stroke survivors with aphasia. *Disability and Rehabilitation*, 40, 1237-1251.

Response: We have revised following your advice.

Reviewer: P5 L16-18 – could be clearer (“the improved dialogue includes a communication...”)

Response: We have revised the sentence.

Reviewer: Some brief details about the co-designers with stroke would be welcome. This has implications for interpreting the unmet needs identified.

Response: We understand the need for details and have added a reference to our publication with details of both the co-design process and all participants.

Reviewer: The 3 areas of unmet need are a little confusing as they each contain a paragraph of content. Could this be more succinct?

Response: We have revised this section.

Reviewer: It is a little unclear how these sessions with people with strokes were co-design sessions as opposed to focus group discussions? It appears that the healthcare professionals were more involved in developing the intervention.

Response: We understand the need for details and have added a reference to our publication with details of both the co-design process and all participants.

Reviewer: P10 L12 – sentence could be split and made clearer.

Response: We have revised this sentence.

Reviewer: Again, it will need to be made clearer earlier in the paper how this was in fact co-design.

Response: We understand the need for details and have added a reference to our publication with details of both the co-design process and all participants.

Reviewer: Once again, you need to detail inclusionary criteria. Will people with aphasia be included, and if so, how will informed consent be obtained? (Similarly, later on, process for conducting communicatively accessible interviews will need to be detailed if so).

Response: We have revised with exclusion criteria.